# Microparticles as BDMDAC (Quaternary Ammonium Compound) Carriers for Water Disinfection: A Layer-by-Layer Approach without Biocide Release

**DOI:** 10.3390/nano13233067

**Published:** 2023-12-02

**Authors:** Marta Redondo, Ana Pereira, Carlos M. Pereira, Luís F. Melo

**Affiliations:** 1LEPABE—Laboratory for Process Engineering, Environment, Biotechnology and Energy, Department of Chemical Engineering, Faculty of Engineering, University of Porto, R. Dr. Roberto Frias, 4200-465 Porto, Portugal; martta.redondo@gmail.com (M.R.); aalex@fe.up.pt (A.P.); 2ALiCE—Associate Laboratory in Chemical Engineering, Faculty of Engineering, University of Porto, R. Dr. Roberto Frias, 4200-465 Porto, Portugal; 3Instituto de Ciências Moleculares/Centro de Investigação em Química da Universidade do Porto (IMS/CIQUP), Faculdade de Ciências da Universidade do Porto, Departamento de Química e Bioquímica, Rua do Campo Alegre 687, 4169-007 Porto, Portugal; cmpereir@fc.up.pt

**Keywords:** layer-by-layer, functionalization, water polishing, hydroxyapatite, calcium carbonate, glass beads

## Abstract

This work studies the antimicrobial activity of benzyldimethyldodecyl ammonium chloride (BDMDAC)-coated microparticles with distinct morphological structures. Functionalized microparticles were prepared by the layer-by-layer (LbL) self-assembly technique on hydroxyapatite (Hap), calcium carbonate (CaCO_3_) and glass beads (GB) cores. All particles were characterized, before and after functionalization, by Fourier-Transform Infrared Spectroscopy (FTIR), Brunner–Emmett–Teller (BET) and Scanning Electron Microscopy (SEM) analyses. Antimicrobial activity was tested against planktonic *Pseudomonas fluorescens*. Planktonic bacteria were exposed to 100 mg/L, 200 mg/L and 400 mg/L of BDMDAC-coated microparticles for 240 min. This strategy promoted a complete bacteria reduction at 200 mg/L for Hap microparticles after 240 min. No release of biocide was detected through HPLC analyses during 2 weeks, suggesting that bacteria inactivation may be attributed to a contact killing mechanism.

## 1. Introduction

It is generally accepted that biocides are important to control microbiological proliferation [1] in engineered systems. However, when dissolved in water, these agents have a potentially negative impact on health and the environment [2]. High levels of consumption of antibiotics and biocides foster the development of microbial resistance, a global health threat [3]. A shift from the development of new antimicrobials to the search of novel nano-microtechnology methods for the prevention of biofilm formation and resistance development has been occurring in the scientific community, since some of these techniques have been proven capable of overcoming the limitations inherent to the current strategies [4]. A significant part of those methods relies on the development of materials with the capacity of repelling and killing pathogens, and are tightly linked to changes in surface chemistry, roughness and charge [5].

After the discovery of antibiotics, there was an explosion in R&D with several antibiotics reaching the market [6]. However, over the past 20 years only two new classes emerged; the reason lies in the resistance development, decreasing the expected profit for pharmaceutical companies [7]. Therefore, new alternatives arose from combining knowledge of engineered systems and antimicrobial materials such as hydrogels, nanoparticles, nanocarbon materials and modified surfaces [8,9,10]. Nanoparticles have been broadly explored and synthesized in this field by using metals such as silver, zinc and titanium, but they mostly work through the slow release of metallic ions to the surrounding medium.

Here, durable and stable antimicrobial particles to minimize biocidal release, bacteria growth and proliferation were prepared for application as water disinfection agents. Different core particles were used as support material for biocide immobilization. Such immobilization aims at creating a dense layer of biocidal molecules on particle surfaces.

Regarding the selection of particles, several features were considered, including size, mechanical properties, chemical modification and price [11]. However, the main concern was with size, from macro- to micro- and nanoscale. While reducing the scale increases the surface–volume ratio allowing higher biocidal loads and higher contact area, for industrial applications, micro and macro particles are preferable to nanoparticles, as they will be easier to retain in open reactor systems and cause lower pressure drops [12]. For the purpose of this work, three distinct core particles were studied: hydroxyapatite (Hap), calcium carbonate (CaCO_3_) and glass beads (GB), their size ranging from 5 to 100 μm. Hap is well known for its great biocompatibility and chemical stability, promoting the development of nano- and microparticles for several applications mainly in the biomedical field [13,14]. In turn, CaCO_3_ particles are also well known for being cheap, porous and made of a very abundant mineral in nature [15]. Glass beads were also used due to their easy chemical modification and stability, mechanical strength, high resistance and low cost [16].

Quaternary ammonium compounds (QACs) were used as biocides; in this case, benzyldimethyldodecyl ammonium chloride (BDMDAC) was selected. QACs are potent cationic surfactants with strong antimicrobial, antifungal and antiviral effects [17]. QACs have been used for a variety of purposes; however, their underlying mechanisms of action are still not fully understood, mainly when immobilized on surfaces. The literature reports that their antimicrobial action is preserved both in solution and when immobilized on surfaces [18]. Some studies reported that not all biocides affect microbial cells in the same way. Barros et al. (2022) showed the difference between the mechanisms of benzalkonium chloride (BAC) and dibromonitrilopropionamide (DBNPA, a non-QAC chemical) activity [19]. While the first one exhibits a lytic action, with high impact on the cell membrane integrity, the second only reveals a moderate electrophilic action. These results confirm that the mechanisms of action of biocides must be considered specifically for each application. Several chemical approaches to modify the particles with QACs were considered, such as graft polymerization [20], layer-by-layer (LbL) [21] assembly and covalent attachment [22]. The positive charge of QACs was crucial for the chemical functionalization here undertaken—the layer-by layer (LbL) technique. LbL is an easy, versatile and cost-effective system that consists in sequentially depositing layers of polyelectrolytes with opposite charges on a core particle, settled on electrostatic interactions, with subsequent incorporation of an active molecule [23].

The antimicrobial efficiency of these particles was assessed against a model bacterium, *Pseudomonas fluorescens*. *Pseudomonas* cover a wide variety of species that can be found in several natural and technological processes, easily forming biofilms on different types of surfaces. *P. fluorescens* forms biofilms with extremely complex, three-dimensional (3D) structures [24,25]. Amongst other processes, quorum sensing and biofilm formation are integral to the various environmental niches occupied by *P. fluorescens* and allow it to colonize surfaces such as hospital equipment and food-grade stainless steel surfaces, as well as plant surfaces, showerheads and even indoor wall surfaces [25,26,27,28]. In addition to this, biofilms are known to be disinfectant-resistant [29]. Therefore, this work aims to explore several core particles as antimicrobial supports to kill planktonic bacteria without biocide release and avoid biofilm formation downstream in technical flow water systems.

## 2. Materials and Methods

### 2.1. Materials

Polyethyleneimine (PEI; molecular weight of 750,000), branched, at 50% (*w*/*v*) in water, poly(sodium 4-styrenesulfonate) (PSS; molecular weight of 70,000) and boric acid were purchased from Sigma-Aldrich (Darmstadt, Germany). Benzyldimethyldodecylammonium chloride (BDMDAC; molecular weight of 339.9) was purchased from Fluka (Tucson, AZ, USA). Hydroxyapatite particles (Hap) of size 5.0 ± 1.0 μm were obtained from Fluidinova S.A, Portugal (nanoXIM-Hap202). Calcium carbonate microparticles (CaCO_3_) with a diameter of 2–4 μm were obtained from PlasmaChem GmbH (Berlin, Germany). All chemicals were used without further purification. Glass beads (GB) with size distribution 100.0 ± 1.0 μm were obtained from Carl Roth GmbH + Co., KG (Stuttgart, Germany).

### 2.2. Layer-by-Layer Assembly

In order to develop antimicrobial particles, three distinct particle cores were functionalized to incorporate BDMDAC. Hydroxyapatite, calcium carbonate and glass beads were used as cores for a layer-by-layer approach (LbL), as schematized in Figure 1. LbL assembly is a safe, economic and environmentally friendly way to modify surfaces. This technique is based on noncovalent interactions, typically electrostatic ones [23]. Two distinct building blocks, oppositely charged, interact with each other through attractive interactions, forming a supramolecular multilayer of polyelectrolytes. Repulsive forces between building blocks of the same type will restrict the layer formed during a single adsorption step [30]. So, LbL consists in the deposition of polyelectrolytes with opposite charges from aqueous solutions, which will subsequently lead to the formation of multilayers.

Hydroxyapatite, calcium carbonate and glass bead particles were firstly dispersed in a polyethyleneimine (PEI) solution (1 mg/mL, 0.1 mol/L borate buffer, pH = 9) for 30 min. The obtained particles were collected, and the excess of PEI was removed by 2 washings with borate buffer. Then, particles interacted with polystyrene sulfonate (PSS) solution (1 mg/mL, 0.1 mol/L borate buffer, pH = 9) for 30 min and the excess of PSS was removed by 2 washings with borate buffer. Borate buffer at pH 9 was chosen to guarantee adequate superficial charge through the whole process. After that, particles were redispersed in BDMDAC solution (1 mg/mL, 0.1 mol/L borate buffer, pH = 9) and mechanically stirred for 30 min, followed by two washes. All procedures are performed at RT. Finally, glass beads were dried in an oven at 80 °C overnight and stored at 4 °C.

In this study, BDMDAC was able to interact electrostatically with the previous layer of PSS (polyanion). In the case of glass bead surface, these were initially activated to expose a hydroxylated surface (exposing –OH groups) [31].

### 2.3. Characterization of Functionalized Particles

#### 2.3.1. Fourier Transform Infrared (FTIR) Analysis

The FTIR spectra were obtained using a Bruker Vector 22 spectrometer in the 400–4000 cm^−1^ region. Microparticle powders before and after LbL were analyzed. For each spectrum, 100 scans were collected with a resolution of 4 cm^−1^ at 25 °C.

#### 2.3.2. Zeta Potential Analysis

The zeta potential (ZP) of the particles was determined by applying an electric field across the particles in solution, using a Nano Zetasizer (Malvern Zetasizer Nano ZS (Malvern Instruments, Works, Malvern, UK) at 25 °C. Microparticles (Hap and CaCO_3_) were suspended in ultrapure water (200 mg/L) by sonication for 10 min in an ultrasonic bath. Then, 1 mL of the suspension was analyzed. ZP was not calculated for GBs dispersions, due to the incapability of the equipment to measure samples with high settling rates.

#### 2.3.3. Scanning Electron Microscopy (SEM) Analysis

Scanning Electron Microscopy (SEM) was performed to analyze the coated particles’ integrity and morphological characteristics. SEM analyses were performed in a Gatan ALTO 2500 model. Samples were prepared by pouring the microparticle powder (Hap, CaCO_3_ and GB) into a thin layer of carbon adhesive tape (conductive material). The excess of microparticles was removed with a stream of pure air.

#### 2.3.4. High-Performance Liquid Chromatography (HPLC)

To quantify the BDMDAC loaded on the surface of coated particles, a high-performance liquid chromatography (HPLC) method was setup. The HPLC system consists of a JASCO PU-2080plus ternary pump, a manual injector equipped with a 20 μL sample loop and a JASCO MD-2015 plus diode array detector. Jasco ChromPass Chromatography data software (version 1.8.6.1) allows the control of the equipment and the data processing. The analytical column was a 100-3 ACE CN (150 mm × 4.6 mm). The gradient begins with mobile phase A (0.025% trifluoroacetic acid in water:acetonitrile (90:10), (*v*/*v*)) and is then transitioned to mobile phase B (0.025% trifluoroacetic acid in water:acetonitrile (39:70, (*v*/*v*)), adapted and applied as described in Santos et al., 2010 [32]. A calibration curve was determined using different BDMDAC concentrations (50, 70, 100, 200, 250 and 300 mg/L) in 0.1 mol/L borate buffer solution at pH 9. All samples injected were dissolved in borate buffer solution.

#### 2.3.5. Brunner–Emmett–Teller (BET) Analysis

Understanding the morphology of the particles is crucial for this work since most properties of the particles are size-dependent. To assess the specific surface area (SBET; m^2^/g), total pore volume, (Vp; cm^3^/g) and pore size distribution, a BET analysis was carried out. The equilibrium isotherms of nitrogen adsorption at −196 °C were determined for all type of particles, before and after functionalization, using a Quantachrome NOVA 4200e equipment. In a first stage, the samples were degassed at 120 °C for 3 h. To texturally characterize the samples, the BET method (SBET) was used by applying the BET equation in the linear area of the graphic, included in the P/P_0_ range of 0.05 to 0.30. The total volume of pores for P/P_0_, which represents the equilibrium pressure divided by the saturation pressure, was determined. The pore size distribution was calculated using the nonlocal density functional theory (NLDFT), available on the software of the equipment, assuming a cylindrical model.

### 2.4. Antimicrobial Activity

#### 2.4.1. Microorganism and Culture Conditions

The *Pseudomonas fluorescens* strain used was isolated in the antimicrobial assays from a drinking water distribution system, previously identified by 16S ribosomal sequence analysis [33]. For achieving optimal growth conditions, the main carbon source is glucose. The optimal temperature is 27 ± 3 °C and the pH is 7. *P. fluorescens* was cryopreserved in a refrigerated chamber at −80 °C, in a mixture of nutrient broth and 30% (*v*/*v*) of glycerol. An inoculum from the cryovial was removed and subsequently distributed evenly over the surface of Plate Count Agar (PCA—Merck, Darmstadt, Germany) and incubated for 24 h at 27 ± 3 °C to allow bacteria propagation.

#### 2.4.2. Biocides

The chosen biocide for this work was benzyldimethyldodecyl ammonium chloride (BDMDAC—Sigma Aldrich), a cationic quaternary ammonium compound (QAC) with a long carbon chain composed of 12 carbons. BDMDAC is a detergent, antibacterial and antiseptic compound. It has been broadly used in eyewashes, nasal sprays and injectable solutions as a preservative and antimicrobial [34].

#### 2.4.3. Killing Assays

The *P. fluorescens* strain and culture conditions were the same as described above. Planktonic cells obtained from liquid medium (composition per liter: 5 g glucose (Fisher Chemical), 2.5 g peptone (Oxoid) and 1.25 g yeast extract (Oxoid) in 0.02 mol/L phosphate buffer pH 7 (KH_2_PO_4_; Na_2_HPO_4_)) were centrifuged and resuspended in a sterile saline solution (0.85% NaCl) to an optical density OD_610nm_ = 0.2 ± 0.02 (bacterial cell counts of approximately 1.5 × 10^8^ CFU/mL). Aliquots were collected to test the antimicrobial effects of the coated particles. Functionalized particles were tested at different concentrations (100, 200 and 400 mg/L) during distinct incubation time points (15, 30, 60, 120 and 240 min at 160 rpm, 25 °C). These concentration values represent the mass of biocide incorporated in the particles divided by the total liquid volume and are obtained by adjusting the total mass of particles used. After each incubation time, particles were allowed to deposit and a 5-fold serial dilution of the microbial suspension (supernatant) was performed followed by the drop plate method to plate the samples on PCA (10 mL of sample for each drop). The number of CFUs was determined after an incubation of 24 h at 30 °C. Control experiments were performed in 0.85% (*v*/*v*) saline solution without particles.

#### 2.4.4. Release Assays

To assess the biocide release from the particles, three independent-release assays were performed, with three replicates for each type of particle tested. Two different BDMDAC concentrations, 300 mg/L and 1000 mg/L, were tested. The corresponding mass of particles was introduced in an Erlenmeyer flask and incubated in 0.1 mol/L borate buffer, pH 9, at 160 rpm. Samples of 1000 μL of the solution were collected at several time points, 15 and 30 min, 1, 2, 4, 6, 24, 48, 72 and 96 h, and 1 and 2 weeks for HPLC analysis. Periodical sampling must not affect the minimum volume necessary to conduct this experiment.

## 3. Results

### 3.1. Microparticle Functionalization

BDMDAC was immobilized in the surface of the three core particles: glass beads (GB), hydroxyapatite (Hap) and calcium carbonate (CaCO_3_), as described in Figure 1. To assess the efficacy of such immobilization, particles were characterized, prior to and after functionalization with the biocide, by different techniques: FTIR, zeta potential and SEM.

#### 3.1.1. FTIR Characterization

To identify the characteristic peaks of the BDMDAC, the FTIR spectrum of the biocide alone was determined and is presented in Figure 2a (grey line). Defined peaks as the stretching vibration of the carbonyl (C=O) group at 1725 cm^−1^, the C-H stretching in the aromatic ring at 2800 cm^−1^ and the N-H stretching near 3300 cm^−1^ were observed.

Calcium carbonate (CaCO_3_) bands may be seen in the FTIR spectra at Figure 2b (purple line) at 1800, 2500 and 2900 cm^−1^. The asymmetric and symmetric stretching of the O-C-O bond correspond to the band at 1600 cm^−1^. The same peaks are observed in CaCO_3_-LbL spectra in Figure 2b (orange line); moreover, the spectral line is sharper at 3540, 2800 and 1600 cm^−1^. However, since CaCO_3_ electrostatically interacts with deposit layers of PEI, PSS and BDMDAC, they might be too weak to be detected by FTIR.

The FTIR spectra for Hap and Hap-LbL are also displayed in Figure 2c. Characteristic peaks of Hap (pink line) were observed, such as CO_3_^2−^ at 1500 cm^−1^, PO_4_^3−^ at 1250 cm^−1^ and HPO_4_^2−^ at 600 cm^−1^. Regarding the Hap-LbL spectrum (blue line), it was not possible to observe the incorporation of amine groups from BDMDAC, but the decrease in the intensity of the PO_4_^3−^ peak at 1250 cm^−1^ corroborates the Hap reaction with polyelectrolyte layers during the LbL procedures. Since less functional groups of PO_4_^3−^ are detected, this means that these are electrostatically linked with the polyelectrolytes (PSS, PEI and BDMDAC).

The GB and GB-LbL spectra displayed in Figure 2d do not show significant differences before (green line) and after the functionalization (black line). Both spectra obtained have low intensity, firstly, because glass is not the best substrate to be examined since it absorbs infrared light and, secondly, due to FTIR limited surface sensitivity. Despite the use of the polyelectrolytes in the LbL functionalization, it was not possible to observe the incorporation of sulfur groups or amide groups, respectively, from polystyrene (around 1000 and 1200 cm^−1^, absorption frequencies of the –SO_3_^−^ moiety) and polyethylenimine (amine groups are typically located at 3400 and 1600 cm^−1^) [22]. The peaks shown in the range of 700 to 1000 cm^−1^ can be attributed to the Si-O-Si symmetric stretching from the soda-lime glass beads.

In short, the FTIR analysis suggests that the Hap-LbL microparticles were functionalized with the BDMDAC but this is not clear for GB-LbL and CaCO_3_-LbL microparticles. This might be associated with a weak electrostatic interaction in the case of CaCO_3_-LbL or due to a technical constraint described above in the case of GB-LbL because of their size, which does not allow an efficient particle functionalization.

#### 3.1.2. Zeta Potential

The success of LbL functionalization with BDMDAC was also evaluated by zeta potential. Regarding CaCO_3_ microparticles, after functionalization, the electrokinetic zeta potential switched from −3.50 mV to 18.10 mV. The electrokinetic zeta potential of Hap changed from −5.58 mV to a 23.45 mV upon functionalization through layer-by-layer assembly. As shown in Figure 1 and, as previously described, microparticle functionalization is accomplished by electrostatic interactions of oppositely charged molecules. BDMDAC, a cationic biocide, is positively charged and shifts the particle surface charge from negative to positive when successfully immobilized. As so, the zeta potential confirms the success of the immobilization reaction. Because the equipment was unable to quantify GBs dispersions, due to their larger size and settling rate, zeta potential was not calculated for these samples.

#### 3.1.3. Surface Characterization (SEM Analysis)

Before and after functionalization, particles were also characterized morphologically by SEM—Figure 3. Their size ranges from 5 µm for hydroxyapatite and calcium carbonate particles to 100 µm for glass beads. Observing Figure 3, it is noted that Hap and CaCO_3_ microparticles (Figure 3a, b) showed rougher surfaces compared to the smoother surfaces of glass beads (Figure 3c). Regarding the effect of surface modification, particles maintained their structural integrity after biocide immobilization. When observing Hap-LbL microparticles, the particles seemed to agglomerate after functionalization (Figure 3a). On the other hand, CaCO_3_-LbL microparticles after treatment showed agglomeration with more polished surfaces (Figure 3b). Glass beads, both before and after LbL assembly, exhibited a smooth and clear surface, spheric shape and homogenous distribution. The results show the effects of immobilization in Hap-LbL and CaCO_3_-LbL microparticles. In contrast, it was not possible to identify any changes in the GB-LbL.

#### 3.1.4. BDMDAC Immobilization

To quantify biocide incorporation into the microparticles, washing solutions of all core modifications were analyzed by HPLC. In the case of Hap and CaCO_3_ microparticles, BDMDAC loading reached 11.34 mg/g of particles and 11.18 mg/g of particles, respectively (Table 1). On the other hand, GBs only incorporated 0.97 mg/g (Table 1). This shows that the higher specific surface area encountered in the case of Hap and CaCO_3_ microparticles, described in Table 1, was advantageous for BDMDAC incorporation.

#### 3.1.5. Surface Properties

To assess the specific surface area and the presence of pores before and after functionalization, a BET analysis was performed. BET is a technique based on the phenomena of adsorption and desorption occurring on the surface of the particles.

Analyzing the BET isotherms and their hysteresis for all the microparticles is crucial to identify the specific shape and structure of the particles’ pores.

The N_2_ isotherms observed for the CaCO_3_ microparticles before and after functionalization are displayed in Figure 4a and Figure 4c, respectively. The profiles obtained did not present any of the six types described in the literature [35]. However, considering the pore size distribution, a mesopore peak appears. The presence of only a few mesopores in CaCO_3_ microparticles corroborated the small surface area obtained.

Regarding the Hap microparticles, they have the same size as the CaCO_3_ microparticles; however, they are structurally different. Analyzing the isotherms obtained in Figure 4b,d, Hap microparticles showed a characteristic profile of isotherms type IV, usually exhibited by mesoporous solids with a cylindrical pore shape [36]. By observing pore size distribution (see data in Appendix A), it was possible to conclude that the pore sizes of Hap microparticles are not only meso but also macro pores.

The summary of the main conclusions gathered form the BET analysis (SBET and total pore volume) is displayed in Table 1.

When looking at the nonfunctionalized particles, it is observed that Hap microparticles have a higher specific surface area, approximately 11 times higher than CaCO_3_ microparticles and 4450 times higher than the glass beads. It is known that, for similar porosities, the specific surface area is inversely proportional to the (square of the) size of the particles [33]. Since glass beads are 20 times larger in diameter and are essentially nonporous, this justifies their low specific surface area determined in the BET analysis and also the lower mass of BDMDAC determined per gram of particle.

Comparing Hap and CaCO_3_ microparticles, Hap has a higher total pore volume (ca. 10 times), conferring a considerably higher specific surface area than the CaCO_3_.

By analyzing each type of particle separately, it is possible to observe a decrease in SBET after applying the LbL technique. This result reinforces the successful functionalization of the particles, since the sequential adhesion of the polyelectrolytes and biocides layers increased the particle size, covering their pores and consequently decreasing the specific surface area.

### 3.2. Biocide Release

Data discussed so far show a successful immobilization of biocide to the core particles of Hap and CaCO_3_. So, it is crucial to understand how strongly BDMDAC is linked to the particle. For that, functionalized particles were placed in aqueous solutions and stirred for two weeks. After each time point, a sample of the aqueous solution was injected into the HPLC and the chromatograms compared to the one from free BDMDAC. Two different concentrations (mass of immobilized biocide/volume of water) were tested—300 mg/L and 1000 mg/L. The HPLC chromatograms were similar for the water exposed and not exposed to the particles. Furthermore, the sampled water did not show any of the characteristic peak observed for the BDMDAC (Figure 2a). This indicates that the microparticles functionalized with BDMDAC through LbL did not release the biocide to the bulk solution to a detectable extent after 2 weeks. This suggests that the coating process was effective and that the biocide adhered successfully, with high stability, to the core particles. Although in certain situations, particularly in the field of medical infections, controlled release is an important mechanism, in environmental aquatic systems, it is desirable to minimize such release, whenever possible, to impede the leaching of a contaminant to the natural water systems and to decrease the risk of creating bacterial resistance towards the released biocide.

### 3.3. Particle Efficacy against Pseudomonas fluorescens

All particles were tested regarding their antimicrobial activity against planktonic *Pseudomonas fluorescens*. Cells were in contact with microparticles for 240 min. at 3 distinct BDMDAC concentrations: 100 mg/L, 200 mg/L and 400 mg/L, and their culturability was evaluated. Results are shown in Figure 5.

When analyzing the killing effect of Hap-LbL and CaCO_3_-LbL microparticles, it is observed that, after 240 min. of contact, both functionalized particles impose a 4 Log reduction for the three tested concentrations (orange and blue lines). The killing kinetics of both microparticles seem similar and concentration-independent. Note that, as we get closer to the limit of detection of cultivable cells, a high standard variation is seen and, in some of the experiments, no colonies were found at all. This only serves to illustrate the uncertainty associated to the lowest values of CFUs, which are close to the detection limit of the culture method and may possibly reach even lower levels.

Surprisingly, the higher number of particles used to immobilize 400 mg/L of biocide did not enhance the antimicrobial activity. The reason behind this might be related with an increase in the agglomeration effect and/or a BDMDAC carbon chain entanglement effect [37], see Section 3.4.2.

Glass beads were tested under the same conditions as the other microparticles, but they did not show bacterial count reduction, which made us consider that the microorganism/biocide interaction is affected by the particles’ morphology, particularly their low specific surface area [31]. When comparing GBs with Hap and CaCO_3_ microparticles, we observed that a 20× decrease in size substantially increased both the immobilization of the biocide, as well as its antimicrobial activity. The effect of particle size has already been widely discussed comparing macro- to nano-scale properties [38]. At the nano scale, materials obtain unique properties and their uptake by biological membranes (bacterial and cytoplasmatic membranes) is easy. In this case, the mere change in scales from quasi-macro (100 micron) to micro size (5 micron) results in a substantial improvement in the antimicrobial activity of the functionalized particle.

### 3.4. Considerations about the Effectiveness of BDMDAC Immobilization

#### 3.4.1. Particle Morphological Characteristics: Amount of Immobilized Biocide vs. Antimicrobial Efficacy

The results discussed so far—schematically presented in Figure 6—show that the morphological properties of the core particles are key to the overall antimicrobial performance of the immobilized biocide, particularly when particle size changes significantly. There are several works reporting that the antimicrobial response depends on intrinsic physicochemical properties of the core material [39,40,41].

Pasquet et al., 2014 [42] reported that highly porous particles are more effective against bacteria and showed that larger pore sizes induce better antimicrobial activity. The presence of meso- and macropores is also known to be associated with high biocidal loadings and highly controlled release systems [43]. However, the present work reveals that these relationships are not so straightforward. In fact, even though Hap-LbL and the CaCO_3_-LbL particles have distinct specific surface areas (SBET), porosities and pore sizes, they are able to incorporate similar amounts of biocide (Table 1) and they show a similar antimicrobial efficacy (Figure 5) against *P. fluorescens*. An acceptable explanation is that the biocide immobilization was concentrated mainly at the outer surfaces of the particles and the effective antimicrobial activity is, therefore, similar for Hap and CaCO_3_ particles since their diameters and external surface areas are similar. This may be the result of the deposition of polyelectrolytes; these molecules may obstruct the pores, inhibiting the further deposition of biocides and reducing the potential surface area of these materials.

#### 3.4.2. Effects of Immobilized Biocide Concentration and of the Particles’ Characteristics

In the previously described antimicrobial assays, a higher concentration of biocide corresponded to a higher mass (and number) of particles in solution, which means more proximity between the microparticles and higher probability of collisions and agglomeration. The agglomeration process is considered an obstacle to the reproducibility of toxicity assays [44]. This effect influences toxicity, affecting cellular uptake, decreasing the effective specific surface area and increasing the sedimentation rate [45,46].

Parallelly to agglomeration, the effect of carbon chain entanglement might also be considered. According to the literature, in n-alkyl-QACs where the n-alkyl chain length is increased beyond 10 carbon units (which is the case of BMDAC), attraction between the adjacent hydrophobic chains extending from the particle’s surface may overcome the electrostatic repulsion of their charged nitrogen head groups [47]. Consequently, when BDMDAC is immobilized in microparticles above a certain surface concentration, the active biocide might be hampered because of this effect, thus hindering the antimicrobial activity. This line of thinking can be applied to the assays at 400 mg/L, where the entanglement effect may indeed be propelled by a high concentration of surface-bound biocide, reducing active biocide availability per unit mass of particles.

Regardless of the antimicrobial efficacy of Hap-LbL and CaCO_3_-LbL particles, it is clear that the underlying mechanism behind bacteria killing was not due to biocidal release from the particles. The most feasible hypothesis is to consider that bacteria become injured upon contact with the immobilized biocide, in a process named contact killing. Contact killing systems, which are based on the use of non-eluting surfaces, are described as possessing a long-term durability, reduced development of antimicrobial resistance and are more environmentally friendly, therefore being considered a sustainable alternative to biocide-leaching approaches [48]. Concerning the practical use of the microparticles with similar antibacterial effect, Hap-LbL are more stable in a wide range of environments, while CaCO_3_-LbL microparticles dissolve at low pH values [49].

#### 3.4.3. Further Research Ahead

It is assumed that immobilized BDMDAC is not readily available as it happens when the same biocide is dissolved in water. This means that free and immobilized concentrations are not comparable per se when assessing the biocidal efficacy and the physical-chemical and biological processes involved.

In fact, the mechanism of antimicrobial particles with incorporated biocidal molecules is far from being completely unveiled, leading to the formulation of different conjectures as to their mode of action [50]. In order to maximize the efficacy of immobilized biocides and to design efficient antimicrobial methods, a deeper understanding of the mechanisms of action and of the subsequent effects on the antimicrobial resistance of bacterial cells is needed [51]. The data on the functionalized particles here presented highlight their potential for applications in the biomedical field and in water treatment and biofilm prevention [10,12,15].

## 4. Conclusions

We successfully incorporated a quaternary ammonium compound (BDMDAC) in three distinct core particles, hydroxyapatite (Hap), calcium carbonate (CaCO_3_) and glass beads (GB) by LbL assembly. Layer-by-layer is a safe, economic and environmentally friendly method. Very importantly, there was no release of biocide from the particles. Afterwards, morphological characterizations were performed and particles with different intrinsic properties as size, surface area, pore size and distribution were tested. Their efficacy against *P. fluorescens* was assessed, the best results having been obtained with Hap-LbL and CaCO_3_-LbL microparticles after a contact time of 240 min, with a 4-log reduction. These particles showed a much higher efficiency than GB-LbL, which is in accordance with their higher specific surface area. The relation between particle concentration and antimicrobial efficacy needs to be more thoroughly assessed, as well as the effects of porosity in these core particles. Research on the effects of immobilized biocides on the acquisition of microbial resistance is the next key step of these studies.

## Figures and Tables

**Figure 1 nanomaterials-13-03067-f001:**
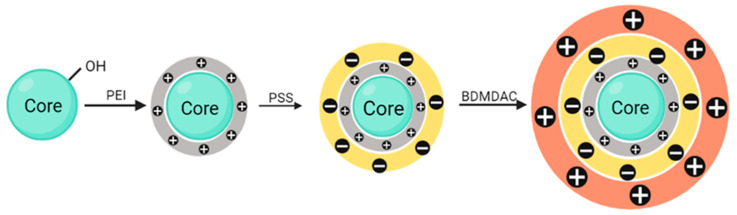
Schematic representation of layer-by-layer (LbL) modification into microparticles (core: hydroxyapatite (Hap) or calcium carbonate (CaCO_3_) or glass beads (GB)). Grey—represents the layer of polycation polyethylenimine (PEI); yellow—represents the layer of polyanion polystyrene sulfonate (PSS); orange—represents the layer of biocide benzyldimethyldodecylammonium chloride (BDMDAC).

**Figure 2 nanomaterials-13-03067-f002:**
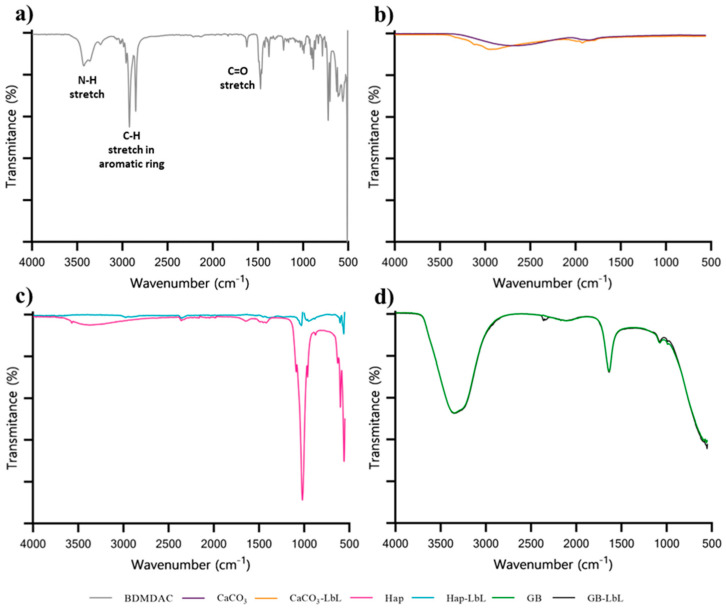
FTIR spectra of (**a**) BDMDAC (grey line); (**b**) CaCO_3_-calcium carbonate microparticles before (purple line) and after functionalization CaCO_3_-LbL (orange line); (**c**) Hap-hydroxyapatite before (pink line) and after functionalization. (**d**) Hap-LbL (blue line) and GB-glass beads before (green line) and after functionalization GB-LbL (black line).

**Figure 3 nanomaterials-13-03067-f003:**
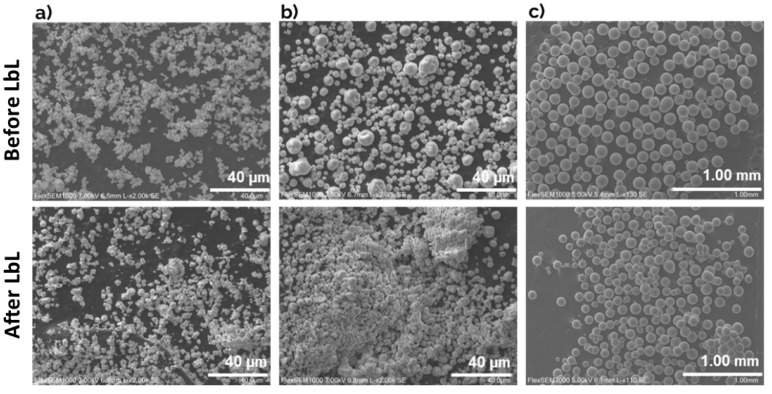
SEM analysis of microparticles before (**upper** line) and after (**bottom** line) layer-by-layer modification: (**a**) CaCO_3_ microparticles, (**b**) Hap microparticles, (**c**) glass beads (GBs).

**Figure 4 nanomaterials-13-03067-f004:**
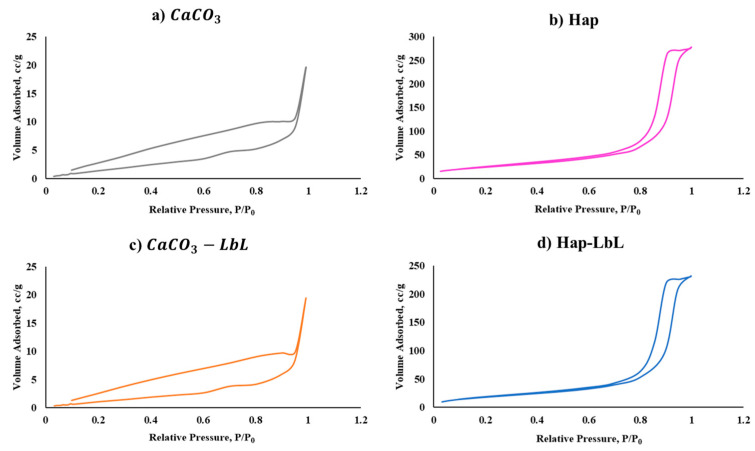
N_2_ isotherms obtained by BET analysis of microparticles before (**a**,**b**) and after (**c**,**d**) layer-by-layer modification, respectively: (**a**,**c**) CaCO_3_ microparticles, (**b**,**d**) Hap microparticles.

**Figure 5 nanomaterials-13-03067-f005:**
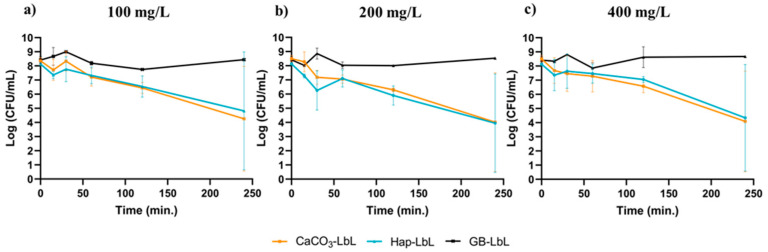
Culturability assay, via log (CFU/mL) counts of *P. fluorescens*, in contact with GB (black line), Hap (blue line) and CaCO_3_ (orange line) coated particles for 3 different BDMDAC concentrations over time.

**Figure 6 nanomaterials-13-03067-f006:**
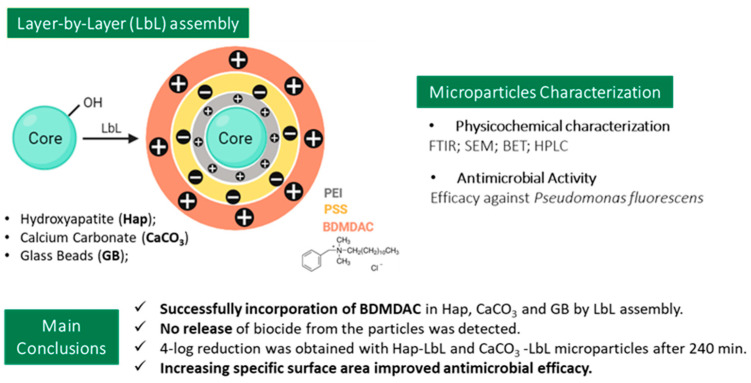
Simplification of the layer-by-layer assembly technique applied to the 3 core particles, showing the opposite charged layers. Analyses were performed to monitor the physicochemical and antimicrobial properties of the particles, before and after functionalization. A summary of the key findings is shown here.

**Table 1 nanomaterials-13-03067-t001:** BET analysis before and after layer-by-layer functionalization and concentration of BDMDAC immobilized into microparticles.

Sample	Size (µm)	S_BET_ (m^2^/g)		Total Pore Volume (m^2^/g)		Mass of 1 g of Particles (g)	Immobilized [BDMDAC] (mg/g)	Nº of Particles for 200 mg/L of BDMDAC
		Before LbL	After LbL	Before LbL	After LbL			
CaCO_3_	2.6 ± 0.3	8	5	0.030	0.030	3.83 × 10^−11^	11.18	4.67 × 10^11^
Hap	5 ± 1.0	89	70	0.429	0.360	2.07 × 10^−10^	11.34	8.53 × 10^10^
GB	100 ± 1.0	0.02	0.02	-	-	1.31 × 10^−6^	0.97	1.58 × 10^8^

## Data Availability

The data presented in this study are available from the corresponding author upon request.

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
