# Peer review of "Microparticles as BDMDAC (Quaternary Ammonium Compound) Carriers for Water Disinfection: A Layer-by-Layer Approach without Biocide Release"

_nanomaterials, 2023, doi:10.3390/nano13233067_

Round 1
Reviewer 1 Report
Comments and Suggestions for Authors
This paper has a certain innovative approach and has currently attracted the attention of many researchers. The antibacterial activity of different BDMDAC coated particles was studied throughout the article. The article also confirmed that layer by layer self-assembly technology is indeed a very effective method for preparing functional particles, and preliminary characterization of these particles was carried out, and their bactericidal mechanism was speculated. But there are still some issues that need to be improved or addressed in this article:
1. There is no reasonable discussion on the formation mechanism of BDMDAC coated particles, and the theoretical basis of the three core particles selected is not fully explained.
2. Some of the literature in the article, especially those involved in the introduction, are not closely related to this article, while others are too outdated.
3. For the prepared functional particles, the determination of their antibacterial activity is indeed effective, but the discussion of their antibacterial mechanism is still in the speculated stage and requires further confirmation by reasonable means.
4. If there are formatting errors such as superscripts and subscripts, as well as language errors in the text, it is also recommended that the author make detailed revisions.
In addition, the description of some details is somewhat controversial, such as even if some antibacterial or metal ions in antibacterial agents may be released into their surrounding environment, it cannot be said that they are completely negative effects. I also hope the author can give detailed consideration.
Reviewer 2 Report
Comments and Suggestions for Authors
The authors immobilized BDMDAC on Hap, CaCO3 and GB microparticles of different sizes using LBL process. Among these, GB microparticles had the largest size of 100 micrometer whilst Hap and CaCO3 exhibited smaller sizes of around 5 and 2-4 micrometers, respectively. All functionalized microparticles were then exposed to P. fluorescens to assess their antimicrobial activity. BDMDAC was poorly immobilized on GB microparticles due to their largest size. Thus GBs had poor antiseptic effect. In contrast, BDMDAC was effectively functionalized on both Hap and porous CaCO3 microparticles leading to their excellent antimicrobial efficacy, i.e. 4-log reduction after 240 min exposure to planktonic bacteria.
In general, I found this article interesting in terms of the use of three kinds of inorganic microparticles of different properties, sizes and biocompatibility by functionalized with cationic BDMDAC for bacteria killing. The experimental was carefully planned and carried out. The results were well discussed and presented. Thus I recommended the publication of this article in its original form.
Reviewer 3 Report
Comments and Suggestions for Authors
I carefully reviewed this manuscript. I smoothly read it.
L49: The term “able” is not required.
L52: “particles’ surface” is changed to “particle surfaces” or “the surfaces of the particles”.
L101: Is the used PEI branched or linear type? I think it is branched one.
L123: The abbreviation of “PEI” should be used.
L241: The abbreviation of chemicals should be used. They are defined on L106-107.
3.2-3.4.3 For readers to understand the results, one or more figures or tables are required.
References
For each reference, all author name should be shown.
